# Scalable Robust Matrix Factorization with Nonconvex Loss

**Quanming Yao**[1,2]**, James T. Kwok**[2]
[1]4Paradigm Inc. Beijing, China
[2]Department of Computer Science and Engineering,
Hong Kong University of Science and Technology, Hong Kong
yaoquanming@4paradigm.com, jamesk@cse.ust.hk

## Abstract

Matrix factorization (MF), which uses the $\ell_2$-loss, and robust matrix factorization (RMF), which uses the $\ell_1$-loss, are sometimes not robust enough for outliers. Moreover, even the state-of-the-art RMF solver (RMF-MM) is slow and cannot utilize data sparsity. In this paper, we propose to improve robustness by using nonconvex loss functions. The resultant optimization problem is difficult. To improve efficiency and scalability, we use majorization-minimization (MM) and optimize the MM surrogate by using the accelerated proximal gradient algorithm on its dual problem. Data sparsity can also be exploited. The resultant algorithm has low time and space complexities, and is guaranteed to converge to a critical point. Extensive experiments show that it outperforms the state-of-the-art in terms of both accuracy and speed.

## 1 Introduction

Matrix factorization (MF) is a fundamental tool in machine learning, and an important component in many applications such as computer vision [1, 38], social networks [37] and recommender systems [30]. The square loss has been commonly used in MF [8, 30]. This implicitly assumes the Gaussian noise, and is sensitive to outliers. Eriksson and van den Hengel [12] proposed robust matrix factorization (RMF), which uses the $\ell_1$-loss, and obtains much better empirical performance. However, the resultant nonconvex nonsmooth optimization problem is much more difficult.

Most RMF solvers are not scalable [6, 12, 22, 27, 40]. The current state-of-the-art solver is RMF-MM [26], which is based on majorization minimization (MM) [20, 24]. In each iteration, a convex nonsmooth surrogate is optimized. RMF-MM is advantageous in that it has theoretical convergence guarantees, and demonstrates fast empirical convergence [26]. However, it cannot utilize data sparsity. This is problematic in applications such as structure from motion [23] and recommender system [30], where the data matrices, though large, are often sparse.

Though the $\ell_1$-loss used in RMF is more robust than the $\ell_2$, still it may not be robust enough for outliers. Recently, better empirical performance is obtained in total-variation image denosing by using the $\ell_0$-loss instead [35], and in sparse coding the capped-$\ell_1$ loss [21]. A similar observation is also made on the $\ell_1$-regularizer in sparse learning and low-rank matrix learning [16, 38, 41]. To alleviate this problem, various nonconvex regularizers have been introduced. Examples include the Geman penalty [14], Laplace penalty [34], log-sum penalty (LSP) [9] minimax concave penalty (MCP) [39], and the smooth-capped-absolute-deviation (SCAD) penalty [13]. These regularizers are similar in shape to Tukey's biweight function in robust statistics [19], which flattens for large values. Empirically, they achieve much better performance than $\ell_1$.

In this paper, we propose to improve the robustness of RMF by using these nonconvex functions (instead of $\ell_1$ or $\ell_2$) as the loss function. The resultant optimization problem is difficult, and existing

RMF solvers cannot be used. As in RMF-MM, we rely on the more flexible MM optimization technique, and a new MM surrogate is proposed. To improve scalability, we transform the surrogate to its dual and then solve it with the accelerated proximal gradient (APG) algorithm [2, 32]. Data sparsity can also be exploited in the design of the APG algorithm. As for its convergence analysis, proof techniques in RMF-MM cannot be used as the loss is no longer convex. Instead, we develop new proof techniques based on the Clarke subdifferential [10], and show that convergence to a critical point can be guaranteed. Extensive experiments on both synthetic and real-world data sets demonstrate superiority of the proposed algorithm over the state-of-the-art in terms of both accuracy and scalability.

**Notation.** For scalar $x$, $\text{sign}(x) = 1$ if $x > 0$, 0 if $x = 0$, and $-1$ otherwise. For a vector $x$, $\text{Diag}(x)$ constructs a diagonal matrix $X$ with $X_{ii} = x_i$. For a matrix $X$, $\|X\|_F = (\sum_{i,j} X_{ij}^2)^{1/2}$ is its Frobenius norm, $\|X\|_1 = \sum_{i,j} |X_{ij}|$ is its $\ell_1$-norm, and $\text{nnz}(X)$ is the number of nonzero elements in $X$. For a square matrix $X$, $\text{tr}(X) = \sum_i X_{ii}$ is its trace. For two matrices $X, Y$, $X \odot Y$ denotes their element-wise product. For a smooth function $f$, $\nabla f$ is its gradient. For a convex $f$, $G \in \partial f(X) = \{U : f(Y) \geq f(X) + \text{tr}(U^\top(Y - X))\}$ is a subgradient.

## 2 Related Work

### 2.1 Majorization Minimization

Majorization minimization (MM) is a general technique to make difficult optimization problems easier [20, 24]. Consider a function $h(X)$ which is hard to optimize. Let the iterate at the $k$th MM iteration be $X^k$. The next iterate is generated as $X^{k+1} = X^k + \arg\min_X f^k(X; X^k)$, where $f^k$ is a surrogate that is being optimized instead of $h$. A good surrogate should have the following properties [24]: (i) $h(X^k + X) \leq f^k(X; X^k)$ for any $X$; (ii) $0 = \arg\min_X (f^k(X; X^k) - h(X^k + X))$ and $h(X^k) = f^k(0; X^k)$; and (iii) $f^k$ is convex on $X$. MM only guarantees that the objectives obtained in successive iterations are non-increasing, but does not guarantee convergence of $X^k$ [20, 24].

### 2.2 Robust Matrix Factorization (RMF)

In matrix factorization (MF), the data matrix $M \in \mathbb{R}^{m \times n}$ is approximated by $UV^\top$, where $U \in \mathbb{R}^{m \times r}$, $V \in \mathbb{R}^{n \times r}$ and $r \ll \min(m, n)$ is the rank. In applications such as structure from motion (SfM) [1] and recommender systems [30], some entries of $M$ may be missing. In general, the MF problem can be formulated as: $\min_{U,V} \frac{1}{2}\|W \odot (M - UV^\top)\|_F^2 + \frac{\lambda}{2}(\|U\|_F^2 + \|V\|_F^2)$, where $W \in \{0, 1\}^{m \times n}$ contains indices to the observed entries in $M$ (with $W_{ij} = 1$ if $M_{ij}$ is observed, and 0 otherwise), and $\lambda \geq 0$ is a regularization parameter. The $\ell_2$-loss is sensitive to outliers. In [11], it is replaced by the $\ell_1$-loss, leading to robust matrix factorization (RMF):

$$\min_{U,V} \|W \odot (M - UV^\top)\|_1 + \frac{\lambda}{2}(\|U\|_F^2 + \|V\|_F^2). \tag{1}$$

Many RMF solvers have been developed [6, 7, 12, 18, 22, 26, 27, 40]. However, as the objective in (1) is neither convex nor smooth, these solvers lack scalability, robustness and/or convergence guarantees. Interested readers are referred to Section 2 of [26] for details.

Recently, the RMF-MM algorithm [26] solves (1) using MM. Let the $k$th iterate be $(U^k, V^k)$. RMF-MM tries to find increments $(\bar{U}, \bar{V})$ that should be added to obtain the target $(U, V)$:

$$U = U^k + \bar{U}, \quad V = V^k + \bar{V}. \tag{2}$$

Substituting into (1), the objective can be rewritten as $H^k(\bar{U}, \bar{V}) \equiv \|W \odot (M - (U^k + \bar{U})(V^k + \bar{V})^\top)\|_1 + \frac{\lambda}{2}\|U^k + \bar{U}\|_F^2 + \frac{\lambda}{2}\|V^k + \bar{V}\|_F^2$. The following Proposition constructs a surrogate $F^k$ of $H^k$ that satisfies properties (i) and (ii) in Section 2.1. Unlike $H^k$, $F^k$ is jointly convex in $(\bar{U}, \bar{V})$.

**Proposition 2.1.** *[26] Let* $\text{nnz}(W_{(i,:)})$ *(resp.* $\text{nnz}(W_{(:,j)})$*) be the number of nonzero elements in the $i$th row (resp. $j$th column) of $W$,* $\Lambda_r = \text{Diag}(\sqrt{\text{nnz}(W_{(1,:)})}, \ldots, \sqrt{\text{nnz}(W_{(m,:)})})$*, and* $\Lambda_c = \text{Diag}(\sqrt{\text{nnz}(W_{(:,1)})}, \ldots, \sqrt{\text{nnz}(W_{(:,n)})})$*. Then,* $H^k(\bar{U}, \bar{V}) \leq F^k(\bar{U}, \bar{V})$*, where*

$$F^k(\bar{U}, \bar{V}) \equiv \|W \odot (M - U^k(V^k)^\top - \bar{U}(V^k)^\top - U^k\bar{V}^\top)\|_1$$
$$+ \frac{\lambda}{2}\|U^k + \bar{U}\|_F^2 + \frac{1}{2}\|\Lambda_r\bar{U}\|_F^2 + \frac{\lambda}{2}\|V^k + \bar{V}\|_F^2 + \frac{1}{2}\|\Lambda_c\bar{V}\|_F^2. \tag{3}$$

*Equality holds iff* $(\bar{U}, \bar{V}) = (0, 0)$.

Because of the coupling of $\bar{U}, V^k$ (resp. $U^k, \bar{V}$) in $\bar{U}(V^k)^\top$ (resp. $U^k \bar{V}^\top$) in (3), $F^k$ is still difficult to optimize. To address this problem, RMF-MM uses the LADMPSAP algorithm [25], which is a multi-block variant of the alternating direction method of multipliers (ADMM) [3].

RMF-MM has a space complexity of $O(mn)$, and a time complexity of $O(mnrIK)$, where $I$ is the number of (inner) LADMPSAP iterations and $K$ is the number of (outer) RMF-MM iterations. These grow linearly with the matrix size, and can be expensive on large data sets. Besides, as discussed in Section 1, the $\ell_1$-loss may still be sensitive to outliers.

## 3 Proposed Algorithm

### 3.1 Use a More Robust Nonconvex Loss

In this paper, we improve robustness of RMF by using a general nonconvex loss instead of the $\ell_1$-loss. Problem (1) is then changed to:

$$\min_{U,V} \dot{H}(U, V) \equiv \sum_{i=1}^{m} \sum_{j=1}^{n} W_{ij} \phi \left( |M_{ij} - [UV^\top]_{ij}| \right) + \frac{\lambda}{2} (\|U\|_F^2 + \|V\|_F^2), \qquad (4)$$

where $\phi$ is nonconvex. We assume the following on $\phi$:

**Assumption 1.** $\phi(\alpha)$ *is concave, smooth and strictly increasing on* $\alpha \geq 0$.

Assumption 1 is satisfied by many nonconvex functions, including the Geman, Laplace and LSP penalties mentioned in Section 1, and slightly modified variants of the MCP and SCAD penalties. Details can be found in Appendix A. Unlike previous papers [16, 38, 41], we use these nonconvex functions as the *loss*, not as regularizer. The $\ell_1$ also satisfies Assumption 1, and thus (4) includes (1).

When the $i$th row of $W$ is zero, the $i$th row of $U$ obtained is zero because of the $\|U\|_F^2$ regularizer. Similarly, when the $i$th column of $W$ is zero, the corresponding column in $V$ is zero. To avoid this trivial solution, we assume the following, as in matrix completion [8] and RMF-MM.

**Assumption 2.** $W$ *has no zero row or column.*

### 3.2 Constructing the Surrogate

Problem (4) is difficult to solve, and existing RMF solvers cannot be used as they rely crucially on the $\ell_1$-norm. In this Section, we use the more flexible MM technique as in RMF-MM. However, its surrogate construction scheme cannot be used here. RMF-MM uses the convex $\ell_1$ loss, and only needs to handle nonconvexity resulting from the product $UV^\top$ in (1). Here, nonconvexity in (4) comes from both from the loss and $UV^\top$.

The following Proposition first obtains a convex upper bound of the nonconvex $\phi$ using Taylor expansion. An illustration is shown in Figure 1. Note that this upper bound is simply a re-weighted $\ell_1$, with scaling factor $\phi'(|\beta|)$ and offset $\phi(|\beta|) - \phi'(|\beta|)|\beta|$. As one may expect, recovery of the $\ell_1$ makes optimization easier. It is known that the LSP, when used as a regularizer, can be interpreted as re-weighted $\ell_1$ regularization [8]. Thus, Proposition 3.1 includes this as a special case.

**Proposition 3.1.** *For any given* $\beta \in \mathbb{R}$, $\phi(|\alpha|) \leq \phi'(|\beta|)|\alpha| + (\phi(|\beta|) - \phi'(|\beta|)|\beta|)$, *and the equality holds iff* $\alpha = \pm\beta$.

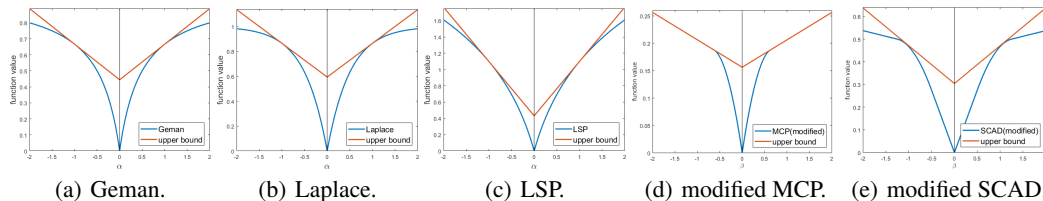

| (a) Geman. | (b) Laplace. | (c) LSP. | (d) modified MCP. | (e) modified SCAD. |

Figure 1: Upper bounds for the various nonconvex penalties (see Table 5 in Appendix A.2) $\beta = 1$, $\theta = 2.5$ for SCAD and $\theta = 0.5$ for the others; and $\delta = 0.05$ for MCP and SCAD.

Given the current iterate $(U^k, V^k)$, we want to find increments $(\bar{U}, \bar{V})$ as in (2). $\dot{H}$ in (4) can be rewritten as: $\dot{H}^k(\bar{U}, \bar{V}) \equiv \sum_{i=1}^m \sum_{j=1}^n W_{ij}\phi(|M_{ij} - [(U^k+\bar{U})(V^k+\bar{V})^\top]_{ij}|) + \frac{\lambda}{2}\|U^k + \bar{U}\|_F^2 + \frac{\lambda}{2}\|V^k + \bar{V}\|_F^2$. Using Proposition 3.1, we obtain the following convex upper bound for $\dot{H}^k$.

**Corollary 3.2.** $\dot{H}^k(\bar{U}, \bar{V}) \le b^k + \frac{\lambda}{2}\|U^k + \bar{U}\|_F^2 + \frac{\lambda}{2}\|V^k + \bar{V}\|_F^2 + \|\dot{W}^k \odot (M - U^k(V^k)^\top - \bar{U}(V^k)^\top - U^k\bar{V}^\top - \bar{U}\bar{V}^\top)\|_1$, where $b^k = \sum_{i=1}^m \sum_{j=1}^n W_{ij}(\phi(|[U^k(V^k)^\top]_{ij}|) - A_{ij}^k|[U^k(V^k)^\top]_{ij}|)$, $\dot{W}^k = A^k \odot W$, and $A_{ij}^k = \phi'(|[U^k(V^k)^\top]_{ij}|)$.

The product $\bar{U}\bar{V}^\top$ still couples $\bar{U}$ and $\bar{V}$ together. As $\dot{H}^k$ is similar to $H^k$ in Section 2.2, one may want to reuse Proposition 2.1. However, Proposition 2.1 holds only when $W$ is a binary matrix, while $\dot{W}^k$ here is real-valued. Let $\Lambda_r^k = \text{Diag}(\sqrt{\text{sum}(\dot{W}_{(1,:)}^k)}, \ldots, \sqrt{\text{sum}(\dot{W}_{(m,:)}^k)})$ and $\Lambda_c^k = \text{Diag}(\sqrt{\text{sum}(\dot{W}_{(:,1)}^k)}, \ldots, \sqrt{\text{sum}(\dot{W}_{(:,n)}^k)})$. The following Proposition shows that $\dot{F}^k(\bar{U}, \bar{V}) \equiv \|\dot{W}^k \odot (M - U^k(V^k)^\top - \bar{U}(V^k)^\top - U^k\bar{V}^\top)\|_1 + \frac{\lambda}{2}\|U^k + \bar{U}\|_F^2 + \frac{1}{2}\|\Lambda_r^k\bar{U}\|_F^2 + \frac{\lambda}{2}\|V^k + \bar{V}\|_F^2 + \frac{1}{2}\|\Lambda_c^k\bar{V}\|_F^2 + b^k$, can be used as a surrogate. Moreover, it can be easily seen that $\dot{F}^k$ qualifies as a good surrogate in Section 2.1: (a) $\dot{H}(\bar{U}+U^k, \bar{V}+V^k) \le \dot{F}^k(\bar{U}, \bar{V})$; (b) $(0,0) = \arg\min_{\bar{U}, \bar{V}} \dot{F}^k(\bar{U}, \bar{V}) - \dot{H}^k(\bar{U}, \bar{V})$ and $\dot{F}^k(0,0) = \dot{H}(0,0)$; and (c) $\dot{F}^k$ is jointly convex in $\bar{U}, \bar{V}$.

**Proposition 3.3.** $\dot{H}^k(\bar{U}, \bar{V}) \le \dot{F}^k(\bar{U}, \bar{V})$, with equality holds iff $(\bar{U}, \bar{V}) = (0,0)$.

**Remark 3.1.** *In the special case where the $\ell_1$-loss is used, $\dot{W}^k = W$, $b^k = 0$ $\Lambda_r^k = \Lambda_r$, and $\Lambda_c^k = \Lambda_c$. The surrogate $\dot{F}^k(\bar{U}, \bar{V})$ then reduces to that in (3), and Proposition 3.3 becomes Proposition 2.1.*

### 3.3 Optimizing the Surrogate via APG on the Dual

LADMPSAP, which is used in RMF-MM, can also be used to optimize $\dot{F}^k$. However, the dual variable in LADMPSAP is a dense matrix, and cannot utilize possible sparsity of $W$. Moreover, LADMPSAP converges at a rate of $O(1/T)$ [25], which is slow. In the following, we propose a time- and space-efficient optimization procedure based on running the accelerated proximal gradient (APG) algorithm on the surrogate optimization problem's dual. Note that while the primal problem has $O(mn)$ variables, the dual problem has only nnz($W$) variables.

**Problem Reformulation**. Let $\Omega \equiv \{(i_1, j_1), \ldots, (i_{\text{nnz}(W)}, j_{\text{nnz}(W)})\}$ be the set containing indices of the observed elements in $W$, $\mathcal{H}_\Omega(\cdot)$ be the linear operator which maps a nnz($W$)-dimensional vector $x$ to the sparse matrix $X \in \mathbb{R}^{m \times n}$ with nonzero positions indicated by $\Omega$ (i.e., $X_{i_t j_t} = x_t$ where $(i_t, j_t)$ is the $t$th element in $\Omega$), and $\mathcal{H}_\Omega^{-1}(\cdot)$ be the inverse operator of $\mathcal{H}_\Omega$.

**Proposition 3.4.** *The dual problem of $\min_{\bar{U}, \bar{V}} \dot{F}^k(\bar{U}, \bar{V})$ is*

$$\min_{x \in \mathcal{W}^k} \mathcal{D}^k(x) \equiv \frac{1}{2}\text{tr}((\mathcal{H}_\Omega(x)V^k - \lambda U^k)^\top A_r^k(\mathcal{H}_\Omega(x)V^k - \lambda U^k)) - \text{tr}(\mathcal{H}_\Omega(x)^\top M)$$

$$+ \frac{1}{2}\text{tr}((\mathcal{H}_\Omega(x)^\top U^k - \lambda V^k)^\top A_c^k(\mathcal{H}_\Omega(x)^\top U^k - \lambda V^k)), \quad (5)$$

*where $\mathcal{W}^k \equiv \{x \in \mathbb{R}^{\text{nnz}(W)} : |x_i| \le [\dot{w}^k]_i^{-1}\}$, $\dot{w}^k = \mathcal{H}_\Omega^{-1}(\dot{W}^k)$, $A_r^k = (\lambda I + (\Lambda_r^k)^2)^{-1}$, and $A_c^k = (\lambda I + (\Lambda_c^k)^2)^{-1}$. From the obtained $x$, the primal $(\bar{U}, \bar{V})$ solution can be recovered as $\bar{U} = A_r^k(\mathcal{H}_\Omega(x)V^k - \lambda U^k)$ and $\bar{V} = A_c^k(\mathcal{H}_\Omega(x)^\top U^k - \lambda V^k)$.*

Problem (5) can be solved by the APG algorithm, which has a convergence rate of $O(1/T^2)$ [2, 32] and is faster than LADMPSAP. As $\mathcal{W}^k$ involves only $\ell_1$ constraints, the proximal step can be easily computed with closed-form (details are in Appendix B.3) and takes only $O(\text{nnz}(W))$ time.

The complete procedure, which will be called Robust Matrix Factorization with Nonconvex Loss (RMFNL) algorithm, is shown in Algorithm 1. The surrogate is optimized via its dual in step 4. The primal solution is recovered in step 5, and $(U^k, V^k)$ are updated in step 6.

**Exploiting Sparsity**. A direct implementation of APG takes $O(mn)$ space and $O(mnr)$ time per iteration. In the following, we show how these can be reduced by exploiting sparsity of $W$.

The objective in (5) involves $A_r^k, A_c^k$ and $\mathcal{W}^k$, which are all related to $\dot{W}^k$. Recall that $\dot{W}^k$ in Corollary 3.2 is sparse (as $W$ is sparse). Thus, by exploiting sparsity, constructing $A_r^k, A_c^k$ and $\mathcal{W}^k$ only take $O(\text{nnz}(W))$ time and space.

---

**Algorithm 1** Robust matrix factorization using nonconvex loss (RMFNL) algorithm.

---
1: initialize $U^1 \in \mathbb{R}^{m \times r}$ and $V^1 \in \mathbb{R}^{m \times r}$;
2: **for** $k = 1, 2, \ldots, K$ **do**
3:     compute $\dot{W}^k$ in Corollary 3.2 (only on the observed positions), and $\Lambda_r^k, \Lambda_c^k$;
4:     compute $x^k = \arg\min_{x \in \mathcal{W}^k} \mathcal{D}^k(x)$ in Proposition 3.4 using APG;
5:     $\bar{U}^k = A_r^k \left( \mathcal{H}_\Omega(x^k) V^k - \lambda U^k \right), \quad \bar{V}^k = A_c^k (\mathcal{H}_\Omega(x^k)^\top U^k - \lambda V^k)$;
6:     $U^{k+1} = U^k + \bar{U}^k, \quad V^{k+1} = V^k + \bar{V}^k$;
7: **end for**
8: **return** $U^{K+1}$ and $V^{K+1}$.

---

In each APG iteration, one has to compute the gradient, objective, and proximal step. First, consider the gradient $\nabla \mathcal{D}^k(x)$ of the objective, which is equal to

$$\mathcal{H}_\Omega^{-1}(A_r^k(\mathcal{H}_\Omega(x)V^k - \lambda U^k)(V^k)^\top) + \mathcal{H}_\Omega^{-1}(U^k[(U^k)^\top \mathcal{H}_\Omega(x) - \lambda(V^k)^\top]A_c^k) - \mathcal{H}_\Omega^{-1}(M). \quad (6)$$

The first term can be rewritten as $\hat{g}^k = \mathcal{H}_\Omega^{-1}(Q^k(V^k)^\top)$, where $Q^k = A_r^k(\mathcal{H}_\Omega(x)V^k - \lambda U^k)$. As $A_r^k$ is diagonal and $\mathcal{H}_\Omega(x)$ is sparse, $Q^k$ can be computed as $A_r^k(\mathcal{H}_\Omega(x)V^k) - \lambda(A_r^k U^k)$ in $O(\text{nnz}(W)r + mr)$ time, where $r$ is the number of columns in $U^k$ and $V^k$. Let the $t$th element in $\Omega$ be $(i_t, j_t)$. By the definition of $\mathcal{H}_\Omega^{-1}(\cdot)$, we have $\hat{g}_t^k = \sum_{q=1}^r Q_{i_t q}^k V_{j_t q}^k$, and this takes $O(\text{nnz}(W)r + mr)$ time. Similarly, computing the second term in (6) takes $O(\text{nnz}(W)r + nr)$ time. Hence, computing $\nabla \mathcal{D}^k(x)$ takes a total of $O(\text{nnz}(W)r + (m+n)r)$ time and $O(\text{nnz}(W) + (m+n)r)$ space (the Algorithm is shown in Appendix B.1). Similarly, the objective can be obtained in $O(\text{nnz}(W)r + (m+n)r)$ time and $O(\text{nnz}(W) + (m+n)r)$ space (details are in Appendix B.2). The proximal step takes $O(\text{nnz}(W))$ time and space, as $x \in \mathbb{R}^{\text{nnz}(W)}$. Thus, by exploiting sparsity, the APG algorithm has a space complexity of $O(\text{nnz}(W) + (m+n)r)$ and iteration time complexity of $O(\text{nnz}(W)r + (m+n)r)$. In comparison, LADMPSAP needs $O(mn)$ space and iteration time complexity of $O(mnr)$. A summary of the complexity results is shown in Figure 2(a).

### 3.4 Convergence Analysis

In this section, we study the convergence of RMFNL. Note that the proof technique in RMF-MM cannot be used, as it relies on convexity of the $\ell_1$-loss while $\phi$ in (4) is nonconvex (in particular, Proposition 1 in [26] fails). Moreover, the proof of RMF-MM uses the subgradient. Here, as $\phi$ is nonconvex, we will use the Clarke subdifferential [10], which generalizes subgradients to nonconvex functions (a brief introduction is in Appendix C). For the iterates $\{X^k\}$ generated by RMF-MM, it is guaranteed to have a *sufficient decrease* on the objective $f$ in the following sense [26]: There exists a constant $\gamma > 0$ such that $f(X^k) - f(X^{k+1}) \geq \gamma \|X^k - X^{k+1}\|_F^2, \forall k$. The following Proposition shows that RMFNL also achieves a sufficient decrease on its objective. Moreover, the $\{(U^k, V^k)\}$ sequence generated is bounded, which has at least one limit point.

**Proposition 3.5.** *For Algorithm 1, $\{(U^k, V^k)\}$ is bounded, and has a sufficient decrease on $\dot{H}$.*

**Theorem 3.6.** *The limit points of the sequence generated by Algorithm 1 are critical points of* (4)*.*

## 4 Experiments

In this section, we compare the proposed RMFNL with state-of-the-art MF algorithms. Experiments are performed on a PC with Intel i7 CPU and 32GB RAM. All the codes are in Matlab, with sparse matrix operations implemented in C++. We use the nonconvex loss functions of LSP, Geman and Laplace in Table 5 of Appendix A, with $\theta = 1$; and fix $\lambda = 20/(m+n)$ in (1) as suggested in [26].

### 4.1 Synthetic Data

We first perform experiments on synthetic data, which is generated as $X = UV^\top$ with $U \in \mathbb{R}^{m \times 5}$, $V \in \mathbb{R}^{m \times 5}$, and $m = \{250, 500, 1000\}$. Elements of $U$ and $V$ are sampled i.i.d. from the standard normal distribution $\mathcal{N}(0, 1)$. This is then corrupted to form $M = X + N + S$, where $N$ is the noise matrix from $\mathcal{N}(0, 0.1)$, and $S$ is a sparse matrix modeling outliers with 5% nonzero elements randomly sampled from $\{\pm 5\}$. We randomly draw $10 \log(m)/m\%$ of the elements from

$M$ as observations, with half of them for training and the other half for validation. The remaining unobserved elements are for testing. Note that the larger the $m$, the sparser is the observed matrix.

The iterate $(U^1, V^1)$ is initialized as Gaussian random matrices, and the iterative procedure is stopped when the relative change in objective values between successive iterations is smaller than $10^{-4}$. For the subproblems in RMF-MM and RMFNL, iteration is stopped when the relative change in objective value is smaller than $10^{-6}$ or a maximum of 300 iterations is used. Rank $r$ is set to the ground truth (i.e., 5). For performance evaluation, we follow [26] and use the (i) testing root mean square error, RMSE $= \sqrt{\|\bar{W} \odot (X - \bar{U}\bar{V}^T)\|_F^2 / \text{nnz}(\bar{W})}$, where $\bar{W}$ is a binary matrix indicating positions of the testing elements; and (ii) CPU time. To reduce statistical variability, results are averaged over five repetitions.

**Solvers for Surrogate Optimization.** Here, we compare three solvers for surrogate optimization in each RMFNL iteration (with the LSP loss and $m = 1000$): (i) LADMPSAP in RMF-MM; (ii) APG(dense), which uses APG but without utilizing data sparsity; and (iii) APG in Algorithm 1, which utilizes data sparsity as in Section 3.3. The APG stepsize is determined by line-search, and adaptive restart is used for further speedup [32]. Figure 2 shows convergence in the first RMFNL iteration (results for the other iterations are similar). As can be seen, LADMPSAP is the slowest w.r.t. the number of iterations, as its convergence rate is inferior to both variants of APG (whose rates are the same). In terms of CPU time, APG is the fastest as it can also utilize data sparsity.

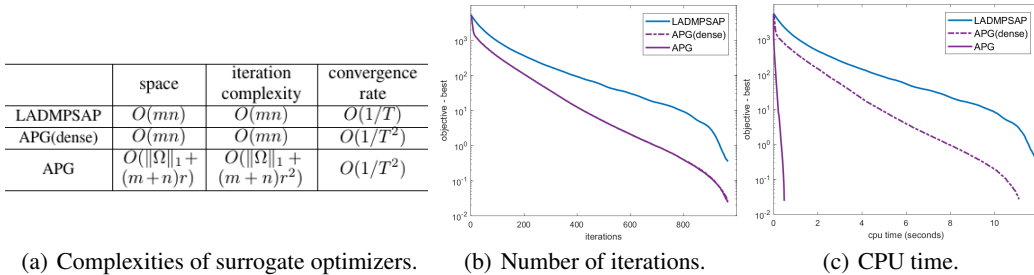

(a) Complexities of surrogate optimizers.  (b) Number of iterations.  (c) CPU time.

Figure 2: Convergence of the objective on the synthetic data set (with the LSP loss and $m = 1000$). Note that the curves for APG-dense and APG overlap in Figure 2(b).

Table 1 shows performance of the whole RMFNL algorithm with different surrogate optimizers.[1] As can be seen, the various nonconvex losses (LSP, Geman and Laplace) lead to similar RMSE's, as has been similarly observed in [16, 38]. Moreover, the different optimizers all obtain the same RMSE. In terms of speed, APG is the fastest, then followed by APG(dense), and LADMPSAP is the slowest. Hence, in the sequel, we will only use APG to optimize the surrogate.

Table 1: Performance of RMFNL with different surrogate optimizers.

| loss | solver | $m = 250$ (nnz: 11.04%) RMSE | CPU time | $m = 500$ (nnz: 6.21%) RMSE | CPU time | $m = 1000$ (nnz: 3.45%) RMSE | CPU time |
|---|---|---|---|---|---|---|---|
| LSP | LADMPSAP | **0.110±0.004** | 17.0±1.4 | **0.072±0.001** | 195.7±34.7 | **0.45±0.007** | 950.8±138.8 |
| | APG(dense) | **0.110±0.004** | 12.1±0.6 | 0.073±0.001 | 114.4±18.8 | **0.45±0.007** | 490.1±91.9 |
| | APG | **0.110±0.004** | **3.2±0.6** | 0.073±0.001 | **5.5±1.0** | **0.45±0.006** | **24.6±3.2** |
| Geman | LADMPSAP | 0.115±0.014 | 20.4±0.8 | 0.074±0.006 | 231.0±36.9 | **0.45±0.007** | 950.8±138.8 |
| | APG(dense) | 0.115±0.011 | 13.9±1.6 | 0.073±0.002 | 146.9±24.8 | **0.45±0.007** | 490.1±91.9 |
| | APG | 0.114±0.009 | **3.1±0.5** | 0.073±0.002 | **8.3±1.1** | **0.45±0.006** | **24.6±3.2** |
| Laplace | LADMPSAP | **0.110±0.004** | 17.1±1.5 | **0.072±0.001** | 203.4±22.7 | **0.45±0.007** | 950.8±138.8 |
| | APG(dense) | **0.110±0.004** | 12.1±2.1 | 0.073±0.003 | 120.9±28.9 | **0.45±0.007** | 490.1±91.9 |
| | APG | **0.111±0.004** | **2.8±0.4** | 0.074±0.001 | **5.6±1.0** | **0.45±0.006** | **24.6±3.2** |

**Comparison with State-of-the-Art Matrix Factorization Algorithms.** The $\ell_2$-loss-based MF algorithms that will be compared include alternating gradient descent (AltGrad) [30], Riemannian preconditioning (RP) [29], scaled alternating steepest descent (ScaledASD) [33], alternative minimization for large scale matrix imputing (ALT-Impute) [17] and online massive dictionary learning (OMDL) [28]. The $\ell_1$-loss-based RMF algorithms being compared include RMF-MM [26], robust matrix completion (RMC) [7] and Grassmannian robust adaptive subspace tracking algorithm

(GRASTA) [18]. Codes are provided by the respective authors. We do not compare with AOPMC [36], which has been shown to be slower than RMC [7].

As can be seen from Table 2, RMFNL produces much lower RMSE than the MF/RMF algorithms, and the RMSEs from different nonconvex losses are similar. AltGrad, RP, ScaledASD, ALT-Impute and OMDL are very fast because they use the simple $\ell_2$ loss. However, their RMSEs are much higher than RMFNL and RMF algorithms. A more detailed convergence comparison is shown in Figure 3. As can be seen, RMF-MM is the slowest. RMFNL with different nonconvex losses have similar convergence behavior, and they all converge to a lower testing RMSE much faster than the others.

Table 2: Performance of the various matrix factorization algorithms on synthetic data.

| loss | algorithm | $m = 250$ (nnz: 11.04%) | | $m = 500$ (nnz: 6.21%) | | $m = 1000$ (nnz: 3.45%) | |
|---|---|---|---|---|---|---|---|
| | | RMSE | CPU time | RMSE | CPU time | RMSE | CPU time |
| $\ell_2$ | AltGrad | 1.062±0.040 | 1.0±0.6 | 0.950±0.005 | 1.8±0.3 | 0.853±0.010 | 6.0±4.2 |
| | RP | 1.048±0.071 | **0.1±0.1** | 0.953±0.012 | 0.4±0.2 | 0.848±0.009 | 1.1±0.1 |
| | ScaledASD | 1.042±0.066 | 0.2±0.1 | 0.950±0.009 | 0.4±0.3 | 0.847±0.009 | 1.2±0.5 |
| | ALT-Impute | 1.030±0.060 | 0.2±0.1 | 0.937±0.010 | 0.3±0.1 | 0.838±0.009 | 1.0±0.2 |
| | OMDL | 1.089±0.055 | **0.1±0.1** | 0.945±0.018 | **0.2±0.1** | 0.847±0.009 | **0.5±0.2** |
| $\ell_1$ | GRASTA | 0.338±0.033 | 1.5±0.1 | 0.306±0.002 | 2.9±0.3 | 0.244±0.009 | 6.1±0.4 |
| | RMC | 0.226±0.040 | 2.8±1.0 | 0.201±0.001 | 2.7±0.5 | 0.195±0.006 | 4.2±2.5 |
| | RMF-MM | 0.194±0.032 | 13.4±0.6 | 0.145±0.009 | 154.9±12.5 | 0.122±0.004 | 827.7±116.3 |
| LSP | RMFNL | **0.110±0.004** | 3.2±0.6 | **0.073±0.001** | 5.5±1.0 | **0.047±0.002** | 14.0±5.2 |
| Geman | RMFNL | 0.114±0.004 | 3.1±0.5 | **0.073±0.001** | 8.3±1.1 | **0.047±0.001** | 19.0±4.9 |
| Laplace | RMFNL | **0.111±0.004** | 2.8±0.4 | **0.074±0.001** | 5.6±1.0 | **0.047±0.002** | 15.9±6.1 |

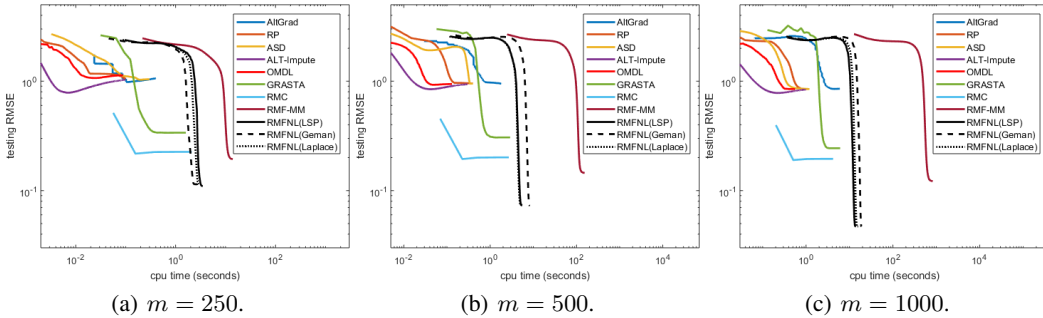

(a) $m = 250$.     (b) $m = 500$.     (c) $m = 1000$.

Figure 3: Convergence of testing RMSE for the various algorithms on synthetic data.

## 4.2 Robust Collaborative Recommendation

In a recommender system, the love/hate attack changes the ratings of selected items to the minimum (hate) or maximum (love) [5]. The love/hate attack is very simple, but can significantly bias overall prediction. As no love/hate attack data sets are publicly available, we follow [5, 31] and manually add permutations. Experiments are performed on the popular *MovieLens* recommender data sets: *MovieLens-100K*, *MovieLens-1M*, and *MovieLens-10M* (Some statistics on these data sets are in Appendix E.1). We randomly select 3% of the items from each data set. For each selected item, all its observed ratings are set to either the minimum or maximum with equal possibilities. 50% of the observed ratings are used for training, 25% for validation, and the rest for testing. Algorithms in Section 4.1 will be compared. To reduce statistical variability, results are averaged over five repetitions. As in Section 4.1, the testing RMSE and CPU time are used for performance evaluation.

Results are shown in Table 3, and Figure 4 shows convergence of the RMSE. Again, RMFNL with different nonconvex losses have similar performance and achieve the lowest RMSE. The MF algorithms are fast, but have high RMSEs. GRASTA is not stable, with large RMSE and variance.

## 4.3 Affine Rigid Structure-from-Motion (SfM)

SfM reconstructs the 3D scene from sparse feature points tracked in $m$ images of a moving camera [23]. Each feature point is projected to every image plane, and is thus represented by a $2m$-dimensional vector. With $n$ feature points, this leads to a $2m \times n$ matrix. Often, this matrix has missing data (e.g., some feature points may not be always visible) and outliers (arising from feature mismatch). We use the Oxford *Dinosaur* sequence, which has 36 images and $4,983$ feature points. As in [26], we extract three data subsets using feature points observed in at least $5, 6$ and $7$

Table 3: Performance on the *MovieLens* data sets. CPU time is in seconds. RMF-MM cannot converge in $10^4$ seconds on the *MovieLens-1M* and *MovieLens-10M* data sets, and thus is not reported.

| loss | algorithm | MovieLens-100K | | MovieLens-1M | | MovieLens-10M | |
|---|---|---|---|---|---|---|---|
| | | RMSE | CPU time | RMSE | CPU time | RMSE | CPU time |
| $\ell_2$ | AltGrad | 0.954±0.004 | 1.0±0.2 | 0.856±0.005 | 30.6±2.5 | 0.872±0.003 | 1130.4±9.6 |
| | RP | 0.968±0.008 | 0.2±0.1 | 0.867±0.002 | 4.4±0.4 | 0.948±0.011 | 199.9±39.0 |
| | ScaledASD | 0.951±0.004 | 0.3±0.1 | 0.878±0.003 | 8.7±0.2 | 0.884±0.001 | 230.2±7.7 |
| | ALT-Impute | 0.942±0.021 | 0.2±0.1 | 0.859±0.001 | 10.7±0.2 | 0.872±0.001 | 198.9±2.6 |
| | OMDL | 0.958±0.003 | **0.1±0.1** | 0.873±0.008 | **2.6±0.5** | 0.881±0.003 | **63.4±4.2** |
| $\ell_1$ | GRASTA | 1.057±0.218 | 4.6±0.3 | 0.842±0.011 | 31.1±0.6 | 0.876±0.047 | 1304.3±18.0 |
| | RMC | 0.920±0.001 | 1.4±0.2 | 0.849±0.001 | 40.6±2.2 | 0.855±0.001 | 526.0±29.5 |
| | RMF-MM | 0.901±0.003 | 402.3±80.0 | — | — | — | — |
| LSP | RMFNL | **0.885±0.006** | 5.9±1.5 | **0.828±0.001** | 34.9±1.0 | **0.817±0.004** | 1508.2±69.1 |
| Geman | RMFNL | **0.885±0.005** | 6.6±1.2 | **0.829±0.005** | 35.3±0.3 | **0.817±0.004** | 1478.5±72.8 |
| Laplace | RMFNL | **0.885±0.005** | 4.9±1.1 | **0.828±0.001** | 35.1±0.2 | **0.817±0.005** | 1513.4±12.2 |

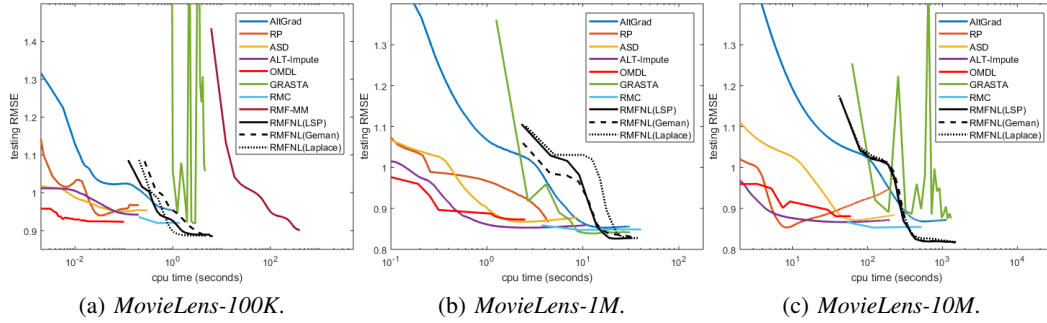

(a) *MovieLens-100K.*    (b) *MovieLens-1M.*    (c) *MovieLens-10M.*

Figure 4: Convergence of testing RMSE on the recommendation data sets.

images. These are denoted "D1" (with size 72×932), "D2" (72×557) and "D3" (72×336). The fully observed data matrix can be recovered by rank-4 matrix factorization [12], and so we set $r = 4$.

We compare RMFNL with RMF-MM and its variant (denoted RMF-MM(heuristic)) in Section 4.2 of [26]. In this variant, the diagonal entries of $\Lambda_r$ and $\Lambda_c$ are initialized with small values and then increased gradually. It is claimed in [26] that this leads to faster convergence. However, our experimental results show that this heuristic leads to more accurate, but not faster, results. Moreover, its key pitfall is that Proposition 2.1 and the convergence guarantee for RMF-MM no longer holds.

For performance evaluation, as there is no ground-truth, we follow [26] and use the (i) mean absolute error (MAE) $\|\bar{W} \odot (\bar{U}\bar{V}^\top - X)\|_1/\text{nnz}(\bar{W})$, where $\bar{U}$ and $\bar{V}$ are outputs from the algorithm, $X$ is the data matrix with observed positions indicated by the binary $\bar{W}$; and (ii) CPU time. As the various nonconvex penalties have been shown to have similar performance, we will only report the LSP here.

Results are shown in Table 4. As can be seen, RMF-MM(heuristic) obtains a lower MAE than RMF-MM, but is still outperformed by RMFNL. RMFNL is the fastest, though the speedup is not as significant as in previous sections. This is because the *Dinosaur* subsets are not very sparse (the percentages of nonzero entries in "D1", "D2" and "D3" are 17.9%, 20.5% and 23.1%, respectively).

Table 4: Performance on the *Dinosaur* data subsets. CPU time is in seconds.

| | D1 | | D2 | | D3 | |
|---|---|---|---|---|---|---|
| | MAE | CPU time | MAE | CPU time | MAE | CPU time |
| RMF-MM(heuristic) | 0.374±0.031 | 43.9±3.3 | 0.381±0.022 | 25.9±3.1 | 0.382±0.034 | 10.8±3.4 |
| RMF-MM | 0.442±0.096 | 26.9±3.4 | 0.458±0.043 | 14.9±2.2 | 0.466±0.072 | 9.2±2.1 |
| RMFNL | **0.323±0.012** | **8.3±1.9** | **0.332±0.005** | **6.8±1.3** | **0.316±0.006** | **3.4±1.0** |

## 5   Conclusion

In this paper, we improved the robustness of matrix factorization by using a nonconvex loss instead of the commonly used (convex) $\ell_1$ and $\ell_2$-losses. Second, we improved its scalability by exploiting data sparsity (which RMF-MM cannot) and using the accelerated proximal gradient algorithm (which is faster than the commonly used ADMM). The space and iteration time complexities are greatly reduced. Theoretical analysis shows that the proposed RMFNL algorithm generates a critical point. Extensive experiments on both synthetic and real-world data sets demonstrate that RMFNL is more accurate and more scalable than the state-of-the-art.

## Footnotes

[1]For all tables in the sequel, the best and comparable results according to the pairwise t-test with 95% confidence are highlighted.

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
