[Supplementary Material]

# A Nonconvex Functions

## A.1 Modification of MCP and SCAD

For the minimax concave penalty (MCP) [39]:

$$\phi(|\alpha|) = \begin{cases} |\alpha| - \frac{\alpha^2}{2\theta} & |\alpha| \le \theta \\ \frac{1}{2}\theta & |\alpha| > \theta \end{cases}.$$

MCP does not meet Assumption 1 as $\phi$ is not strictly increasing when $|\alpha| > \theta$. To avoid this problem, we can modify its $\phi$ as $\tilde{\phi}(|\alpha|) = \phi(|\alpha|) + \delta|\alpha|$, where $\delta > 0$ is a small constant (Figure 1(d)). The smoothly clipped absolute deviation (SCAD) penalty [13] can be modified in the same way (Figure 1(e)).

## A.2 Definitions

Table 5 shows the nonconvex functions that can be used by RMFML.

Table 5: Example nonconvex regularizers ($\theta > 2$ for SCAD and $\theta > 0$ for others is a constant). Here, $\delta > 0$ is a small constant to ensure that the $\phi$'s for MCP and SCAD are strictly increasing.

| | $\phi(|\alpha|)$ |
|---|---|
| Geman penalty | $\frac{|\alpha|}{\theta + |\alpha|}$ |
| Laplace penalty | $1 - \exp\left(-\frac{|\alpha|}{\theta}\right)$ |
| log-sum-penalty (LSP) | $\log\left(1 + \frac{|\alpha|}{\theta}\right)$ |
| minimax concave penalty (MCP) | $\begin{cases} (1+\delta)|\alpha| - \frac{\alpha^2}{2\theta} & \alpha \le \theta \\ \frac{1}{2}\theta^2 + \delta|\alpha| & \alpha > \theta \end{cases}$ |
| smoothly clipped absolute deviation (SCAD) penalty | $\begin{cases} (1+\delta)|\alpha| & |\alpha| \le 1 \\ \frac{-\alpha^2 + 2\theta|\alpha| - 1}{2(\theta-1)} + \delta|\alpha| & 1 < |\alpha| \le \theta \\ \frac{(1+\theta)}{2} + \delta|\alpha| & |\alpha| > \theta \end{cases}$ |

# B Details of the APG Algorithm

## B.1 Computing the Gradient

The complete procedure for computing the gradient is shown in Algorithm 2.

---

**Algorithm 2** Computing $\nabla \mathcal{D}^k(x)$ by exploiting sparsity.

1: set $X_{i_t j_t} = x_t$ for all $(i_t, j_t) \in \Omega$; // i.e., $X = \mathcal{H}_\Omega(x)$
2: $Q^k = A_r^k(XV^k) - \lambda(A_r^k U^k)$;
3: obtain $\hat{g}^k \in \mathbb{R}^{\text{nnz}(W)}$ with $\hat{g}_t^k = \sum_{q=1}^r Q_{i_t q}^k V_{j_t q}^k$;
4: $P^k = A_c^k(X^\top U^k) - \lambda(A_c^k V^k)$;
5: obtain $\breve{g}^k \in \mathbb{R}^{\text{nnz}(W)}$ with $\breve{g}_t^k = \sum_{q=1}^r U_{i_t q}^k P_{j_t q}^k$;  // i.e., $\breve{g}^k = \mathcal{H}_\Omega^{-1}(U^k(P^k)^\top)$
6: **return** $\hat{g}^k + \breve{g}^k - \mathcal{H}_\Omega^{-1}(M)$.

---

## B.2 Computing the Objective

By the definition of $\mathcal{H}_\Omega(x)$, we construct a sparse matrix $X = \mathcal{H}_\Omega(x)$. We then compute the first term in (5) as $\frac{1}{2}\|P^k \sqrt{A_r^k}\|_F^2$ where $P^k = XV^k - \lambda U^k$. Note that $X$ is sparse with $O(\text{nnz}(W))$ nonzero elements and $A_r^k$ is a diagonal, the computation of the first term in (5) takes $O(\text{nnz}(W)r + mr)$ time, where $r$ is the number of columns in $U^k$. Let $y = \mathcal{H}_\Omega^{-1}(M)$. The second term in (5) can then be computed as $\sum_{i=1}^{\text{nnz}(W)} x_i y_i$, which takes $O(\text{nnz}(W))$ time. For the last term in (5), it can be computed similarly as the first term in $O(\text{nnz}(W)r + nr)$ time. Moreover, we can see that only $O(\text{nnz}(W) + (m+n)r)$ space is needed.

The whole procedure for computing the objective is shown in Algorithm 3. It takes a total of $O(\text{nnz}(W) + (m+n)r)$ space and $O(\text{nnz}(W)r + (m+n)r)$ time.

---

**Algorithm 3** Computing $\mathcal{D}^k(x)$ by exploiting sparsity.

---

1: set $X_{i_t j_t} = x_t$ for all $(i_t, j_t) \in \Omega$;
2: $a_1 = \frac{1}{2}\|\sqrt{A_r^k} P^k\|_F^2$ where $P^k = XV^k - \lambda U^k$;
3: $a_2 = \frac{1}{2}\|\sqrt{A_c^k} Q^k\|_F^2$ where $Q^k = X^\top U^k - \lambda V^k$;
4: $a_3 = \sum_{i=1}^{\text{nnz}(W)} x_i y_i$ where $y = \mathcal{H}_\Omega^{-1}(M)$;
5: **return** $a_1 + a_2 + a_3$.

---

### B.3 Computing the Proximal Step

For the proximal step with (5), a closed-form solution can be obtained by the following Lemma.

**Lemma B.1** ([4]). *For any given $z$, $x^* = \arg\min_{x \in \mathcal{W}^k} \frac{1}{2}\|x - z\|_F^2 = [\text{sign}\,(z_i)\min(|z_i|, (\dot{w}_i^k)^{-1})]$.*

## C   Clarke Subdifferential

We first introduce two definitions from [10].

**Definition C.1** (Clarke subdifferential). *Let $f : \mathbb{R}^{m \times n} \to \mathbb{R}$ be a locally Lipschitz function.[2] The* Clarke generalized directional derivative *of $f$ at $X$ in the direction of $V$ is:*

$$f^\circ(X, V) \equiv \limsup_{Y \to X, \lambda \to 0} \frac{1}{\lambda}[f(Y + \lambda V) - f(Y)].$$

*The* Clarke subdifferential *of $f$ at $X$ is*

$$\partial^\circ f(X) \equiv \{\xi : f^\circ(X, V) \geq \text{tr}(\xi^\top V), \forall V \in \mathbb{R}^{m \times n}\}.$$

Note that $f$ in Definition C.1 can be neither convex nor smooth.

**Definition C.2** (Critical point). *A point $X$ is a critical point of $f$ if it satisfies $0 \in \partial^\circ f(X)$.*

## D   Proofs

### D.1   Preliminaries

In the section, we first introduce some Lemmas that will be used later in the proof.

The critical points for problem (4) are defined in the following Lemma.

**Lemma D.1.** *Let $C = M - UV^\top$. $(U, V)$ is a critical point of (4) if $0 \in (W \odot S)V + \lambda U$ and $0 \in (W \odot S)^\top U + \lambda V$, where $S_{ij} = \text{sign}\,(C_{ij})\,\phi'(|C_{ij}|)$ if $C_{ij} \neq 0$, and $S_{ij} \in [-\phi'(0), \phi'(0)]$ otherwise.*

*Proof.* For a nonconvex penalty function $\phi$ satisfying Assumption 1, from Proposition 5 in [15], its Clark subdifferential is

$$\begin{cases} \partial^\circ \phi(|\alpha|) = \text{sign}\,(\alpha) \cdot \phi'(|\alpha|) & \text{if } \alpha \neq 0 \\ \partial^\circ \phi(|\alpha|) \in [-\phi'(0), \phi'(0)] & \text{otherwise} \end{cases}. \tag{7}$$

By Definition C.2, if $(U, V)$ is a critical point of (4), it satisfies

$$(0, 0) \in \partial^\circ \dot{H}(U, V). \tag{8}$$

Combining (7) and (8), we obtain the Lemma.  □

**Lemma D.2.** *Define the row sum* $\text{sum}(\dot{W}^k_{(i,:)}) = \sum_{j=1}^n \dot{W}^k_{ij}$, *and the column sum* $\text{sum}(\dot{W}^k_{(:,j)}) = \sum_{i=1}^m \dot{W}^k_{ij}$. *Then,* $\|\dot{W}^k \odot (\bar{U}\bar{V}^\top)\|_1 \le \frac{1}{2}\|\Lambda^k_r \bar{U}\|_F^2 + \frac{1}{2}\|\Lambda^k_c \bar{V}\|_F^2$, *where*

$$\Lambda^k_r = \text{Diag}(\sqrt{\text{sum}(\dot{W}^k_{(1,:)})}, \dots, \sqrt{\text{sum}(\dot{W}^k_{(m,:)})}),$$

*and*

$$\Lambda^k_c = \text{Diag}(\sqrt{\text{sum}(\dot{W}^k_{(:,1)})}, \dots, \sqrt{\text{sum}(\dot{W}^k_{(:,n)})}).$$

*Equality holds iff* $(\bar{U}, \bar{V}) = (0,0)$.

*Proof.* First, we have

$$\|\dot{W}^k \odot (\bar{U}\bar{V}^\top)\|_1 = \left\| \dot{W}^k \odot \begin{bmatrix} \underline{u}_1^\top \underline{v}_1 \cdots \underline{u}_1^\top \underline{v}_n \\ \cdots \\ \underline{u}_m^\top \underline{v}_1 \cdots \underline{u}_m^\top \underline{v}_n \end{bmatrix} \right\|_1$$

$$= \sum_{i=1}^m \sum_{i=1}^n \dot{W}^k_{ij} \left| \underline{u}_i^\top \underline{v}_j \right|, \tag{9}$$

where $\underline{u}_i$ is the $i$th row in $\bar{U}$ (similar, for $\underline{v}_j$ in $\bar{V}$). From the Cauchy inequality, we have

$$\left| \underline{u}_i^\top \underline{v}_j \right| \le \|\underline{u}_i\|_2 \|\underline{v}_j\|_2 \le \frac{1}{2} \left( \|\underline{u}_i\|_2^2 + \|\underline{v}_j\|_2^2 \right).$$

Together with (9), we have

$$\|\dot{W}^k \odot (\bar{U}\bar{V}^\top)\|_1 \le \frac{1}{2} \sum_{i=1}^m \sum_{j=1}^n \dot{W}^k_{ij} \left( \|\underline{u}_i\|_2^2 + \|\underline{v}_j\|_2^2 \right) = \frac{1}{2}\|\Lambda_r \bar{U}\|_F^2 + \frac{1}{2}\|\Lambda_c \bar{V}\|_F^2,$$

and the equality holds only when $(\bar{U}, \bar{V}) = (\mathbf{0}, \mathbf{0})$. $\qquad\square$

## D.2  Proposition 3.1

*Proof.* Note that $\phi(x)$ is concave on $x \ge 0$. For any $y \ge 0$, we have
$$\phi(y) \le \phi(x) + (y - x)\phi'(x).$$
Let $y = |\beta|$ and $x = |\alpha|$. We obtain
$$\phi(|\beta|) \le \phi(|\alpha|) + (|\beta| - |\alpha|)\phi'(|\alpha|).$$
As $\phi$ is concave and strictly increasing on $\mathbb{R}^+$, equality holds iff $\beta = \pm\alpha$. $\qquad\square$

## D.3  Corollary 3.2

*Proof.* This Corollary can be easily obtained (i) using Proposition 3.1 on the nonconvex loss in (4); and (ii) $U = U^k + \bar{U}$ and $V = V^k + \bar{V}$. $\qquad\square$

## D.4  Proposition 3.3

*Proof.* From the Cauchy inequality, we have
$$\|\dot{W}^k \odot (M - (U^k + \bar{U})(V^k + \bar{V})^\top)\|_1 \tag{10}$$
$$\le \|\dot{W}^k \odot (M - U^k(V^k)^\top - \bar{U}(V^k)^\top - U^k\bar{V}^\top)\|_1 + \|\dot{W}^k \odot (\bar{U}\bar{V}^\top)\|_1.$$
For the last term, using Lemma D.2, we have
$$\|\dot{W}^k \odot (\bar{U}\bar{V}^\top)\|_1 \le \frac{1}{2} \left( \|\Lambda^k_r \bar{U}\|_F^2 + \|\Lambda^k_c \bar{V}\|_F^2 \right). \tag{11}$$
Combining (10) and (11), we have

$$\sum_{i=1}^m \sum_{j=1}^n \dot{W}^k_{ij} \phi\left( \left| M_{ij} - [UV^\top]_{ij} \right| \right) \le \|\dot{W}^k \odot (M - (U^k + \bar{U})(V^k + \bar{V})^\top)\|_1$$

$$+ \frac{1}{2} \left( \|\Lambda^k_r \bar{U}\|_F^2 + \|\Lambda^k_c \bar{V}\|_F^2 \right) + b^k. \tag{12}$$

Adding $\frac{\lambda}{2}\|U^k + \bar{U}\|_F^2 + \frac{\lambda}{2}\|V^k + \bar{V}\|_F^2$ to both side of (12), we obtain the Proposition.

Besides, from Lemma D.2, the equality in the Proposition holds only when $(\bar{U}, \bar{V}) = (0,0)$. $\qquad\square$

## D.5 Proposition 3.4

*Proof.* Using the fact that $\|X\|_1 = \max_{\|Y\|_\infty \leq 1} \text{tr}(X^\top Y)$ [4], where $\|Y\|_\infty = \max_{i,j} |Y_{ij}|$ is the $\ell_\infty$-norm, $\mathcal{D}^k(x)$ can be rewritten as

$$\max_{x \in \mathcal{W}^k} \min_{\bar{U}, \bar{V}} \mathcal{P}(x, \bar{U}, \bar{V}),$$

where

$$\mathcal{P}(x, \bar{U}, \bar{V}) \equiv \text{tr}(\mathcal{H}_\Omega(x)^\top (M - \bar{U}(V^k)^\top - U^k \bar{V}^\top)) \tag{13}$$
$$+ \frac{\lambda}{2}\|U^k + \bar{U}\|_F^2 + \frac{1}{2}\|\Lambda_r^k \bar{U}\|_F^2 + \frac{\lambda}{2}\|V^k + \bar{V}\|_F^2 + \frac{1}{2}\|\Lambda_c^k \bar{V}\|_F^2.$$

As (13) is an unconstrained, smooth and convex problem on $\bar{U}$, the optimal solution is obtained when $\nabla_{\bar{U}} \mathcal{P}(X, \bar{U}, \bar{V}) = 0$. Then,

$$\bar{U} = A_r^k (\mathcal{H}_\Omega(x) V^k - \lambda U^k). \tag{14}$$

Similarly, we obtain

$$\bar{V} = A_c^k (\mathcal{H}_\Omega(x)^\top U^k - \lambda V^k). \tag{15}$$

Substituting (14) and (15) back into (13), we obtain $\mathcal{D}^k(X)$ in the Proposition. $\qquad \square$

## D.6 Proposition 3.5

First, Proposition 3.5 can be written as follows.

**Proposition D.3.** *For Algorithm 1,*

(i). $\{(U^k, V^k)\}$ *is bounded.*

(ii). $\{(U^k, V^k)\}$ *has a sufficient decrease on* $\dot{H}$, *i.e.,* $\dot{H}(U^k, V^k) - \dot{H}(U^{k+1}, V^{k+1}) \geq \gamma \|U^{k+1} - U^k\|_F^2 + \gamma \|V^{k+1} - V^k\|_F^2$, *where* $\gamma > 0$ *is a constant; and*

(iii). $\lim_{k \to \infty}(U^{k+1} - U^k) = 0$ *and* $\lim_{k \to \infty}(V^{k+1} - V^k) = 0$.

*Proof.* First note that

$$\inf_{U, V} H(U, V) \geq 0, \quad \lim_{\substack{\|U\|_F \to \infty \\ \|V\|_F \to \infty}} H(U, V) = \infty. \tag{16}$$

Then, the sequence $\{U^k\}$ and $\{V^k\}$ is bounded, and we obtain the result in part (i).

Thus, there exists a positive constant $c$ such that

$$c_1 \geq |[U^k(V^k)^\top]_{ij}|, \quad \forall i, j, k.$$

From Assumption 1, $\phi$ is a strictly increasing function, thus $\phi' > 0$. Then, there exists a positive constant $c_2$ such that

$$\phi'\left(|[U^k(V^k)^\top]_{ij}|\right) \geq c_2 \equiv \phi'(c_1).$$

From Assumption 2, each row and column in $W$ has at least one nonzero element. By the definition of $\Lambda_r^k$ in Proposition 3.3, its diagonal elements are given by

$$\left[\Lambda_r^k\right]_{ii} \geq \sqrt{\sum_{j=1}^n W_{ij} c_2}.$$

The same holds for $\Lambda_c^k$. Thus, there exists a constant $\alpha > 0$ such that all diagonal elements in $\Lambda_r^k$ and $\Lambda_c^k$ are not smaller than $\alpha$.

As $(\bar{U}^k, \bar{V}^k)$ is the optimal solution of $\min \dot{F}^k$, then

$$(0, 0) \in \partial \dot{F}^k \left(\bar{U}^k, \bar{V}^k\right). \tag{17}$$

Define

$$\dot{J}^k(\bar{U}, \bar{V}) \equiv \|\dot{W}^k \odot (M - U^k(V^k)^\top - \bar{U}(V^k)^\top - U^k\bar{V}^\top)\|_1 + \frac{\lambda}{2}\|U^k + \bar{U}\|_F^2 + \frac{\lambda}{2}\|V^k + \bar{V}\|_F^2 + b^k.$$

Recall the definition of $\dot{F}^k$. From (17), we have

$$(G_{\bar{U}^k}, G_{\bar{V}^k}) \in \partial \bar{J}^k(\bar{U}^k, \bar{V}^k).$$

Thus,

$$(0,0) = (G_{\bar{U}^k}, G_{\bar{V}^k}) + \left((\Lambda_r^k)^2\bar{U}, (\Lambda_c^k)^2\bar{V}\right). \tag{18}$$

Multiplying $(\bar{U}^k, \bar{V}^k)$ on both side of (18), we have

$$0 = \text{tr}(G_{\bar{U}^k}^\top \bar{U}^k) + \text{tr}(G_{\bar{V}^k}^\top \bar{V}^k) + \|(\Lambda_r^k)^2\bar{U}\|_F^2 + \|(\Lambda_c^k)^2\bar{V}\|_F^2. \tag{19}$$

As $\dot{J}^k$ is a convex function, by the definition of the subgradient, we have

$$\dot{J}^k(0,0) \geq \dot{J}^k(\bar{U}^k, \bar{V}^k) - \text{tr}(G_{\bar{U}^k}^\top \bar{U}^k) - \text{tr}(G_{\bar{V}^k}^\top \bar{V}^k). \tag{20}$$

Combining (19) and (20), we obtain

$$\dot{J}^k(0,0) \geq \dot{J}^k(\bar{U}^k, \bar{V}^k) + \|(\Lambda_r^k)^2\bar{U}\|_F^2 + \|(\Lambda_c^k)^2\bar{V}\|_F^2$$

$$\geq \dot{H}^k(\bar{U}^k, \bar{V}^k) + \frac{1}{2}\|(\Lambda_r^k)^2\bar{U}\|_F^2 + \frac{1}{2}\|(\Lambda_c^k)^2\bar{V}\|_F^2. \tag{21}$$

Note that

$$\dot{J}^k(\mathbf{0}, \mathbf{0}) = H(U^k, V^k),$$
$$\dot{H}^k(\bar{U}^k, \bar{V}^k) = H(U^{k+1}, V^{k+1}),$$

and using (21), we have

$$H(U^k, V^k) - H(U^{k+1}, V^{k+1}) \geq \frac{1}{2}\|\Lambda_r^k\bar{U}^k\|_F^2 + \frac{1}{2}\|(\Lambda_c^k)^2\bar{V}^k\|_F^2 \geq \frac{\alpha}{2}\left(\|\bar{U}^k\|_F^2 + \|\bar{V}^k\|_F^2\right). \tag{22}$$

Thus, we obtain the result in part (ii) in Proposition 3.5 (with $\gamma = \alpha/2$).

Summing all inequalities in (22) from $k = 1$ to $K$, we have

$$H(U^1, V^1) - H(U^{K+1}, V^{K+1}) \geq \sum_{k=1}^{K} \frac{\alpha}{2}\|\bar{U}^k\|_F^2 + \frac{\alpha}{2}\|\bar{V}^k\|_F^2.$$

From (16), we have

$$\sum_{k=1}^{\infty} \|\bar{U}^k\|_F^2 < \infty, \sum_{k=1}^{\infty} \|\bar{V}^k\|_F^2 < \infty, \tag{23}$$

which indicates that

$$\lim_{k\to\infty} \|\bar{U}^k\|_F^2 = \lim_{k\to\infty} \|(U^k - U^{k+1})\|_F^2 = 0,$$
$$\lim_{k\to\infty} \|\bar{V}^k\|_F^2 = \lim_{k\to\infty} \|(V^k - V^{k+1})\|_F^2 = 0.$$

Then, we have the result in part (iii). $\qquad\qquad\qquad\qquad\qquad\qquad\qquad\qquad\square$

### D.7 Proposition D.4

The following connects the subgradient of surrogate $\dot{F}^k$ to the Clarke subdifferential of $\dot{H}$.

**Proposition D.4.** *(i)* $\partial \dot{F}^k(0,0) = \partial^\circ \dot{H}^k(0,0)$*; (ii) If* $0 \in \partial^\circ \dot{H}^k(0,0)$*, then* $(U^k, V^k)$ *is a critical point of* (4)*.*

*Proof.* **Part (i).** We prove this by the Clark subdifferential of $\dot{H}^k$ and subgradient of $\dot{F}^k$.

- Clark subdifferential of $\dot{H}^k$: Let $C^H = M - UV^\top$. By the definition of Clark differential, we have

$$\partial_U^\circ \dot{H}^k(\bar{U}, \bar{V}) = (W \odot S^H)(V^k + \bar{V}) + \lambda(U^k + \bar{U}), \qquad (24)$$

$$\partial_V^\circ \dot{H}^k(\bar{U}, \bar{V}) = (W \odot S^H)^\top(U^k + \bar{U}) + \lambda(V^k + \bar{V}), \qquad (25)$$

where $S_{ij}^H = \text{sign}\left(C_{ij}^H\right) \cdot \phi'\left(\left|C_{ij}^H\right|\right)$ if $C_{ij}^H \neq 0$, and $S_{ij}^H \in [-\phi'(0), \phi'(0)]$ otherwise.

- Subgradient of $\dot{F}^k$: Let $C^F = M - U^k(V^k)^\top - \bar{U}^k(V^k)^\top - U^k(\bar{V}^k)^\top$. For $\dot{F}^k$, we have

$$\partial_U \dot{F}^k(\bar{U}, \bar{V}) = (\dot{W}^k \odot S^F)(V^k + \bar{V}^k) + \lambda(U^k + \bar{U}) + (\Lambda_r^k)^2 \bar{U}, \qquad (26)$$

$$\partial_V \dot{F}^k(\bar{U}, \bar{V}) = (\dot{W}^k \odot S^F)^\top(U^k + \bar{U}^k) + \lambda(V^k + \bar{V}) + (\Lambda_c^k)^2 \bar{V}, \qquad (27)$$

where $S_{ij}^F = \text{sign}\left(C_{ij}^F\right)$ if $C_{ij}^F \neq 0$, and $S_{ij}^F \in [-1, 1]$ otherwise.

Note that when $\bar{U} = 0$ and $\bar{V} = 0$, we have $C^H = C^F$. By the definition of $\dot{W}^k = A^k \odot W$, we also have $W \odot S^H = \dot{W}^k \odot S^F$. Finally, the last term in (26) vanishes to zero as $\bar{U} = 0$. Thus, (24) is exactly the same as (26). Similarly (25) is also the same as (27). As a result, we have $\partial^\circ \dot{F}^k(\mathbf{0}, \mathbf{0}) = \partial^\circ \dot{H}^k(0, 0)$.

**Part (ii).** From the definition of $\dot{H}$ in (4) and $\dot{H}^k$ in Proposition 3.3, we have

$$\dot{H}^k(\bar{U}, \bar{V}) = \dot{H}(U^k + \bar{U}, V^k + \bar{V}).$$

Thus, if $(\mathbf{0}, \mathbf{0}) \in \partial^\circ \dot{H}^k(\mathbf{0}, \mathbf{0})$, we have

$$(0, 0) \in \partial^\circ \dot{H}(U^k, V^k),$$

which shows that $(U^k, V^k)$ is a critical point. $\qquad\square$

### D.8   Theorem 3.6

*Proof.* From Proposition 3.5, we know that there is at least one limit point for the sequence $\left\{(U^k, V^k)\right\}$. Let $\left\{(U^{k_j}, V^{k_j})\right\}$ be one of its subsequences, and

$$U^* = \lim_{k_j \to \infty} U^{k_j}, \quad V^* = \lim_{k_j \to \infty} V^{k_j},$$

where $(U^*, V^*)$ is a limit point. Using Proposition D.4, we have

$$\lim_{k_j \to \infty} \partial^\circ \dot{F}^{k_j}\left(\bar{U}_{k_j}, \bar{V}_{k_j}\right) = \lim_{k_j \to \infty} \partial^\circ \dot{F}^{k_j}(0, 0) = \lim_{k_j \to \infty} \partial^\circ \dot{H}^{k_j}(0, 0) = \partial^\circ \dot{H}(U^*, V^*).$$

Thus, $(0, 0) \in \partial^\circ \dot{H}(U^*, V^*)$, which shows that $(U^*, V^*)$ is a critical point (Lemma D.1). $\qquad\square$

## E   Additional Materials for the Experiments

### E.1   Statistics of *MovieLens*.

The statistics of *MovieLens* data sets are in following Table 6.

Table 6: *MovieLens* data sets used.

|  | number of users | number of movies | number of ratings | % nonzero elements |
|---|---|---|---|---|
| *MovieLens-100K* | 943 | 1,682 | 100,000 | 6.30 |
| *MovieLens-1M* | 6,040 | 3,449 | 999,714 | 4.80 |
| *MovieLens-10M* | 69,878 | 10,677 | 10,000,054 | 1.34 |

## Footnotes

[2]A function is called locally Lipschitz continuous if for every $X$ in its domain there exists a neighborhood $\mathcal{U}$ of $X$ such that $f$ restricted to $\mathcal{U}$ is Lipschitz continuous.