[Reviews · NeurIPS 2018]

Reviewer 1



General comments ------------- The paper is well written and propose a well motivated, although incremental, idea for improving robustness of matrix factorization with unknown values. Details -------- Model + convergence ================== - The paper would be much easier to read with less inline equation, in particular in Proposition/Corollary. I think it would be beneficial to drop the \dot on the proposed algorithm, as they are not essential (you may just override the definitions provided for l1 loss) - Although sparsyfying regularization are mentionned in the introduction, they are left undiscussed in the rest of the paper. Can the proposed algorithm accomodate eg l1 penalty on the term V ? - The proposed algorithm involve APG in an inner loop: what is the criterion to stop this inner loop ? Even though the convergence rate of APG is better than ADMM, this is not guaranteed to provide faster convergence for the entire algorithm. This should be discussed more. Experiments ----------- The only experiment performed on a dataset of reasonable size (Movielens 10M) shows bad performance: a well tuned l2 matrix factorization algorithm tuned with eg SGD or coordinate descent, performs under 0.79 RMSE on this dataset (see, among many, Mensch et al., Dictionary Learning for Massive Matrix Factorization). As such, the interest of robust non-convex losses is hardly clear on a real dataset --- and the baselines reported in the paper are wrong. Of course, RMFNL does perform better on datasets generated to require robustness during fitting. However, this is hardly a valid argument to promote the use of new losses.

Reviewer 2



In this paper, an algorithm for matrix factorization (MF) is proposed, which provide better robustness to outlier’s behavior compared to state of the art algorithms for Robust MF (RMF). Another nice property of the new algorithm is that it is suitable for sparse data matrices which is a clear advantage when it is compared against previous approaches to RMF. Basically, the authors propose to replace the l1 penalty with a nonconvex regularizer and provide results for various selections of regularizers. Since the optimization in this case is not simple, the authors propose to use a Majorization Minimization (MM) optimization approach with a definition of a new MM surrogate. The paper provides a nice introduction to the problem of robust matrix factorization, providing a complete overview of the state of the art and giving good motivations for the development of a new algorithm. I found the paper well written, technically sounded and having a nice structure. The theoretical treatment of the results is adequate giving guarantees of convergence. The paper includes a rich experimental section with useful comparisons with previously proposed algorithms and evaluating several options for regularizers. I definitively recommend the acceptance of this submission for presentation in NIPS. However, I found few minor issues which need attention for the final submission. Issues: • Font sizes in all figures are too small. Please consider resizing figures. • In section 4.1 Synthetic Data, outliers are generated by using constant values {+5, -5} in the 5% of the entries. I think, it would be very useful to analyze the effect of varying the percentage of affected entries (instead of a fixed 5%) and also to consider other values for outliers with some probabilistic distribution. I think that, by adding these new analysis the applicability of the algorithm would be better assessed. The authors agreed to add these additional results to the final version.

Reviewer 3



This paper proposes a type of matrix factorization by replacing nuclear norm with it's equivalent Frobius norm and considered seeveral nonconvex loss functions. It then derive a computationing method to solve the system. The proposed method is run on several data sets to show it's validity. Strengths. It consider a set of loss functions such as Gemem, Laplace, etc. It derive several theoretical results. Extensive experients on several data sets. Weakness. The proposed method is a slight variant of the widely studied nuclear norm regularized formulation such a RPCA. Since nuclear norm || Z||_* = min {UV=Z} (||U||^2 + ||V||^2)/2 [Straingely, this well-known formula is not explained in this highly math paper!!!] Thus the paper has very little novalty. The weighting W has also appeared in many research papers. Although experiment show some results, the paper did not compare with many similar formulations using nuclear norm, and also the reasons of the differences.